# Old but Gold: The Surgeon's Affair to Manage Inguinal Hernia

Mario Giuffrida [1,2,*] , Gabriela Elisa Nita [1] and Federico Biolchini [1]

1    General Surgery Unit, Department of Surgery, Azienda USL–IRCCS di Reggio Emilia,
     42123 Reggio Emilia, Italy
2    General Surgery Unit, Department of Medicine and Surgery, Parma University Hospital, 43121 Parma, Italy
*    Correspondence: mario.giuffrida4@gmail.com

**Abstract:** Purpose: Inguinal hernia repair is a common surgical procedure. It was widely reported worldwide during the COVID-19 pandemic. To manage the lack of anesthesiologists, we have introduced a new protocol to manage inguinal hernia repair. Methods: This protocol is the result of a strong collaboration between surgeons and anesthesiologists. It was based on EHS recommendations and the well-described percutaneous ilioinguinal–iliohypogastric and genitofemoral nerves block. Results: More than 400 patients have been treated at our institution. The application of the protocol has led to a sensible reduction in initially planned spinal anesthesia. The complications traditionally related to spinal anesthesia have not been reported in 80% of the patients. Only three patients required the infusion of atropine or flumazenil without the need to involve anesthesiologist. Conclusion: The application of our protocol seems promising. Preliminary results have shown the safety and efficacy of percutaneous ilioinguinal–iliohypogastric and genitofemoral nerves block. The combination of this kind of anesthesia with wound protector and adequate postoperative pain control can lead to a reproducible system avoiding the not strictly necessary presence of an anesthesiologist. The changes that have occurred in the healthcare system in recent years should be new opportunities for the improvement of resources and results.

**Keywords:** inguinal hernia; anesthesia; pain; surgery; hernia repair

## 1. Introduction and Rationale

The COVID-19 pandemic brings anesthesiologists and intensive care physicians to the mainstay of clinical workload and healthcare managements' focus. A shortage of anesthesiologists has been widely reported worldwide. Anesthesiologists play a fundamental role in surgery.

The shortage of physicians is reflected on the long waiting times to access health services, especially for non-urgent pathologies [1].

Inguinal hernia repair is a common surgical procedure. The lifetime risk of developing inguinal hernia is approximately 25% for men [2–4].

Anesthesia for inguinal hernia depends on many factors, including the type, size, and location of the hernia. The different levels of anesthesia include: local anesthesia that theoretically can be performed in the surgeon's office with just a nurse in attendance; monitored anesthesia care with low levels of sedation, which can be raised to a higher level if the patient does not tolerate the discomfort or the surgeon needs to extend the length of the incision; general anesthesia is the last option.

The European Hernia Society (EHS) recommends traditional local anesthetic wound infiltration as the main anesthetic option to perform inguinal hernia repair, avoiding, when possible, spinal anesthesia with high doses of long-acting anesthetics. Excluding young anxious patients, morbid obesity and incarcerated hernia EHS recommend that patients with ASA 3 or 4 can also benefit from local anesthesia [5].

Inguinal hernia repair is accompanied by acute-to-chronic postoperative discomfort. Postoperative pain is associated with many negative outcomes, such as fear, anxiety,

patient discomfort, cardiovascular events, pulmonary atelectasis, poor wound healing, and ventilation problems, which can lead to postoperative complications, delayed rehabilitation, and a reduced level of function and quality of life [6–14].

The lack of anesthesiologists and the burden of inguinal hernias are the main problems that we had to fight against.

To manage the burden of inguinal hernia repair and increase the number of anesthesiologists, we adopted a new protocol based on the not novel and well-described percutaneous ilioinguinal–iliohypogastric and genitofemoral nerves block.

The Ilioinguinal–iliohypogastric and genitofemoral nerve block has been considered a suitable anesthetic method for both children and adult patient populations during surgical procedures in the inguinal region, mainly hernia repairs. The technique is also used for postoperative analgesia after general anesthesia surgical procedures. Additionally, this nerve block may be an effective solution for the treatment of chronic pain after inguinal hernia surgery.

## 2. Methods

In the early 2022, after the COVID-19 pandemic, due to the increased number of patients on waiting lists for inguinal hernia repair and the concomitant reduction in available anesthesiologists, as they were occupied with COVID-19 pandemic and oncologic surgery responsibilities, we decided to adopt this new protocol to overcome the difficulties that the COVID-19 pandemic has led to in our daily practice. After revising the literature and a strong cooperation between surgeons and anesthesiologists, we decided to start this new protocol for the management of inguinal hernia repair using a ilioinguinal–iliohypogastric and genitofemoral nerves block.

## 3. Preoperative Evaluation

This protocol is the result of a strong collaboration between surgeons and anesthesiologists. The application of this protocol must meet inclusion criteria.

The inclusion criteria are assessed during the first medical evaluation.

The following conditions/factors are assessed: past medical history, allergy, and concurrent medications.

The collected information guides the surgeon on choosing the type of anesthesia and assessing the need of preoperative exams. Preoperative blood exams are performed only in ASA 2–3 patients; preoperative electrocardiogram (ECG) is performed only in selected cases to assess the chances of experiencing a heart-related problem during surgery.

Every patient on the waiting list had been reevaluated for their inclusion in this protocol. Every patient who met the inclusion criteria had undergone open inguinal hernia repair.

The inclusion criteria for receiving a percutaneous ilioinguinal–iliohypogastric and genitofemoral nerves block include:

- Patients with inguinal hernia;
- ASA score of <3;
- Aged 18 years or older;
- Patients without mental disorders;
- Patients without complicated hernia.

  Exclusion criteria:

- ASA score of 3 with uncontrolled disease;
- ASA score of >4;
- BMI > 35;
- Patients with severe psychiatric disorders;
- Pediatric patients;
- Late elderly with mental health issues;
- Complicated hernia (incarcerated or strangulated hernia).

## 4. Operative Room Setup

The healthcare staff is composed of two surgeons and two nurses. The first surgeon is the team leader; he leads the nurse anesthesiologist during anesthesia and during the administration of other medications.

The basic equipment used to perform percutaneous nerve block includes an ultrasound machine with a linear or curved transducers.

Medications used for anesthesia are summarized in Table 1.

**Table 1.** Medications required for anesthesia and pain relief.

| Medications | Dilution | Initial Dose | Stage of Anesthesia and Pain Control |
|---|---|---|---|
| Fentanyl 50 micrograms/mL | None | 50 micrograms IV direct administered slowly over 1–2 min. | Induction |
| Midazolam 1 mg/mL | 10 mL 0.9% Sodium Chloride | 1 mL IV | Before surgical incision |
| Mepivacaine 10 mg/mL–Ropivacaine 7.5 mg/mL | 100 mL 0.9 % Sodium Chloride | | Ilioinguinal–iliohypogastric and genitofemoral nerve block |
| Mepivacaine 10 mg/mL–Ropivacaine 7.5 mg/mL | 100 mL 0.9 % Sodium Chloride | 20 mL | Wound infiltration |
| Atropine 0.1 mg/mL | None | | If necessary |
| Flumazenil 0.1 mg/mL | None | | If necessary |
| Naloxone 0.4 mg/mL | 10 mL 0.9% Sodium Chloride | | If necessary |
| Ketorolac 30 mg/mL | 100 mL 0.9 % Sodium Chloride | IV | After surgery |

## 5. Anesthesia

### 5.1. Intravenous Sedation

The use of intravenous sedation increases the acceptability of local anesthesia techniques. Moreover, it improves the success rate of the repair. Fentanyl and midazolam sedation provides rapid recovery after hernia repair, guarantying a better pain control during the surgical procedure.

### 5.2. Ilioinguinal–Iliohypogastric Nerves Block

The patient lies in a supine position. The ilioinguinal–iliohypogastric nerves block is performed unilaterally using ultrasound guidance and placed in the transversus abdominis plane. Ilio-hypogastric and ilioinguinal nerves are the terminal branches of the anterior rami of the L1 spinal nerve. They emerge from the upper part of the lateral border of the psoas major muscle; both nerves cross obliquely anterior to the quadratus lumborum and iliacus muscles and perforate the transverse abdominis muscle near the anterior part of the iliac crest. In the anterior abdominal trunk, the nerves travel between the transverse abdominis and the internal oblique muscles [15].

A high-frequency, linear, high-resolution probe is initially kept perpendicularly on the lateral abdominal wall at the midaxillary line between the anterior superior iliac spine and the navel. In this place, the three abdominal muscles are seen below the subcutaneous fat and the plane between the internal oblique, and the transversus abdominis muscle is identified. The peritoneum can be seen as the fascia layer underneath the transversus abdominis muscle. Between the layers of the transversus abdominis and the internal oblique muscle, the splitting of the fascia layer is usually observed. It is on this plane where the ilioinguinal and iliohypogastric nerves pass through. Sometimes, both nerves pierce the internal oblique and appear between the internal and external oblique muscles.

The needle is inserted in the plane in a medial-to-lateral direction; 10 mL of anesthetic (Mepivacaine 10 mg/mL–Ropivacaine 7.5 mg/mL) is injected in the transversus abdominis plane, as seen in Figures 1–4.

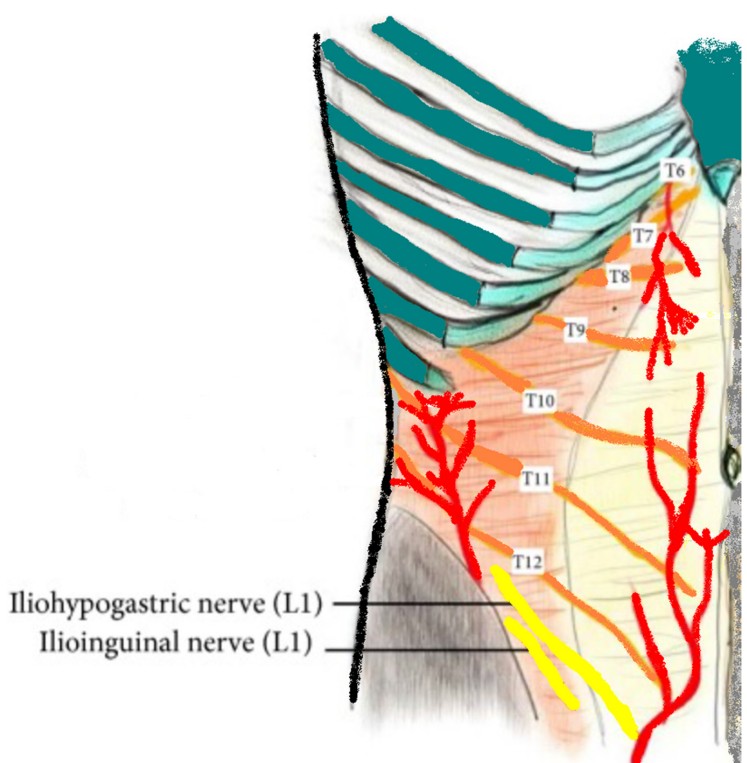

**Figure 1.** Ilioinguinal–iliohypogastric nerves anatomy; T6–T12, thoracoabdominal nerves.

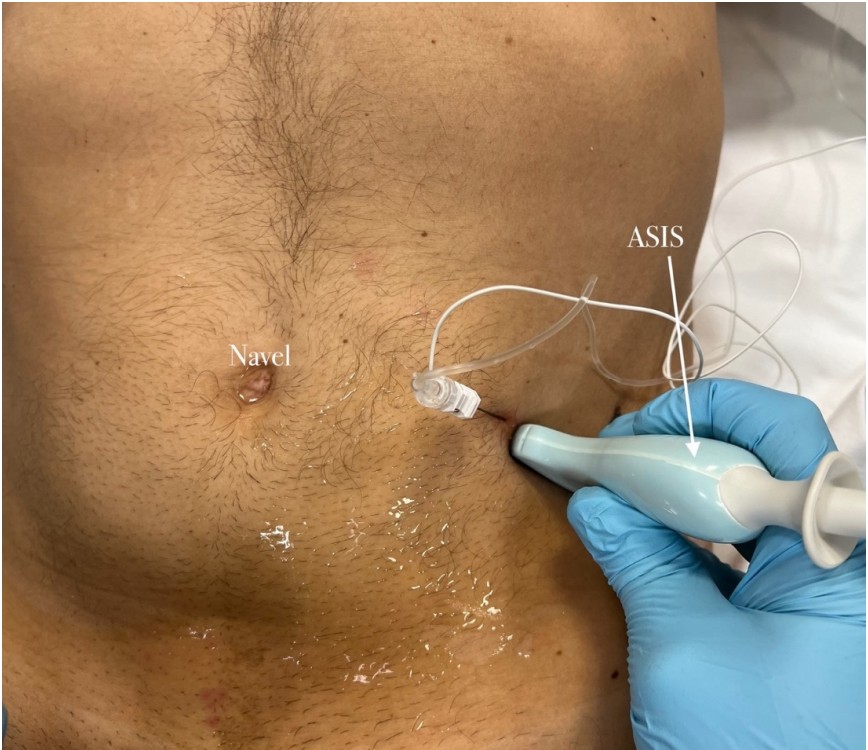

**Figure 2.** Probe position.

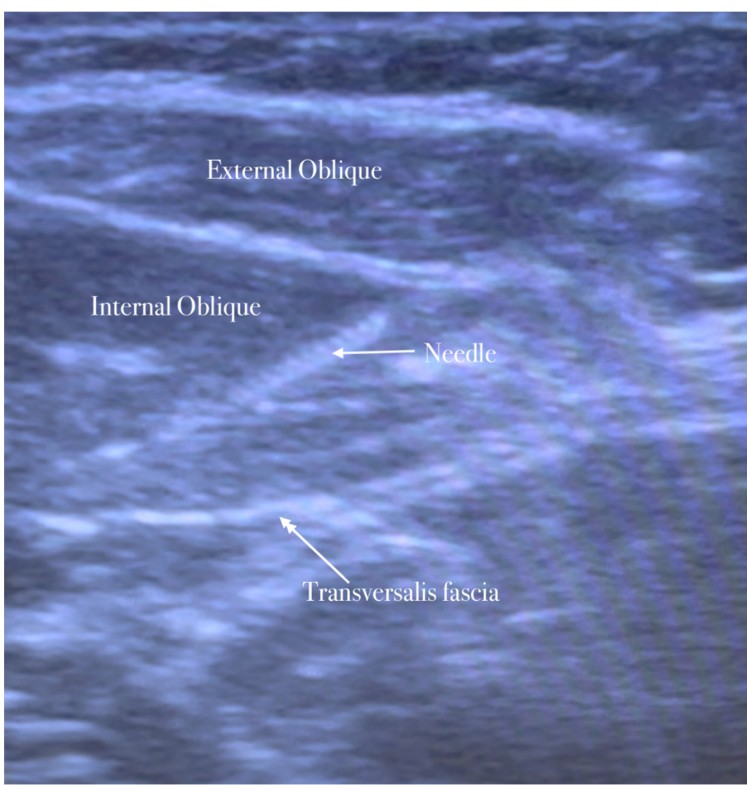

**Figure 3.** The needle is inserted through the three abdominal muscles in the transversus abdominis plane.

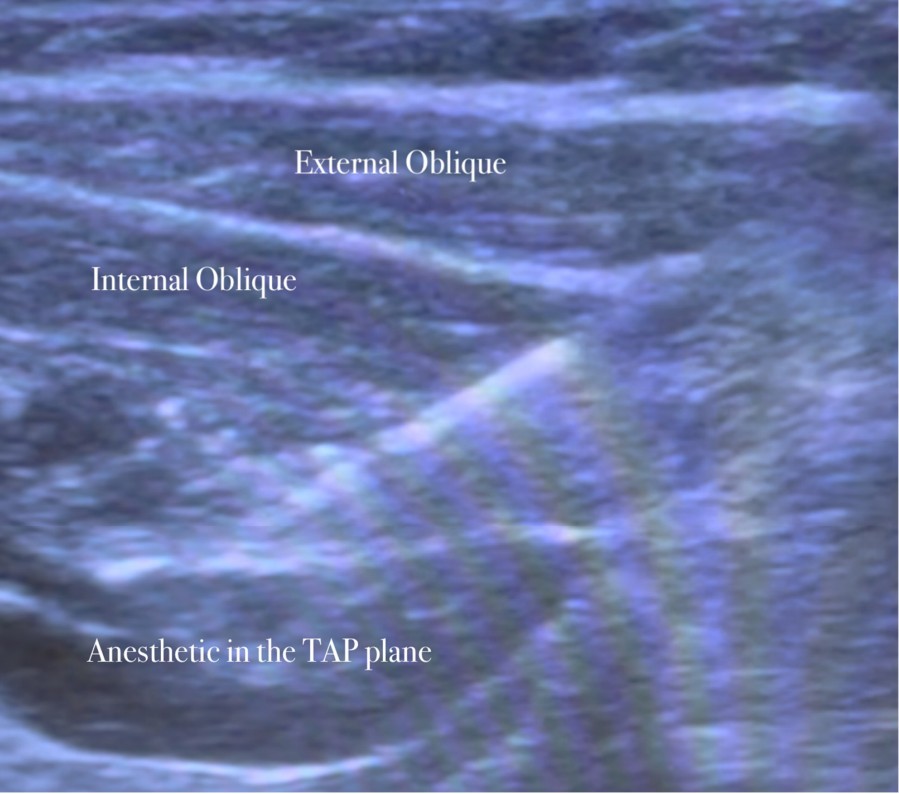

**Figure 4.** The anesthetic is injected in the transversus abdominis plane.

*5.3. Genitofemoral Nerve Block*

The second step of anesthesia consists of the genitofemoral nerve block. The genitofemoral nerve block decreases the pain induced by the traction of the hernia sac and improves the quality of analgesia for surgery in the inguinal region. The procedure is similar to the ilioinguinal–iliohypogastric nerves block. The genitofemoral nerve is formed from the first and second ventral rami of the lumbar nerve. It emerges on the anterior surface of the psoas major along the medial border, descends on the psoas major within the fascia iliaca, and crosses the posterior to the ureter and peritoneum. The nerve follows the lateral border of the common and external iliac artery. It is divided into genital and femoral branches above the inguinal ligament. The genital branch of the genitofemoral nerve passes through the transversalis and spermatic fascia before it enters the deep inguinal ring. It lies immediately laterally or deeply in the spermatic cord/round ligament and supplies the cremaster muscle [16,17].

The high-frequency linear probe is initially kept perpendicular to the inguinal ligament just above the femoral vessels. The final position of the probe is about 2 cm lateral to the pubic tubercle. In this position, the femoral artery is identified. The inguinal canal lies above and medially the femoral artery appearing as an oval or circular structure. The probe is then moved slightly in the medial direction away from the femoral artery.

The needle is inserted in the plane in a medial-to-lateral direction; 10 mL of anesthetic is injected in the transversus abdominis plane, as seen in Figures 5–7.

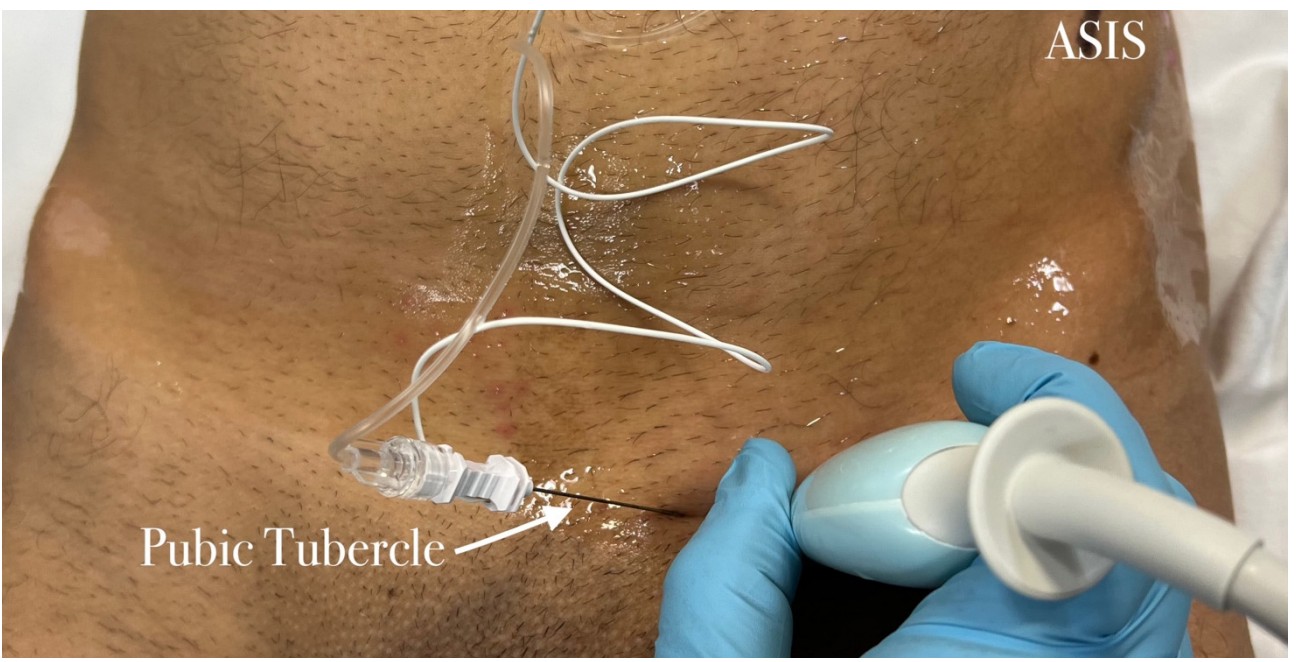

**Figure 5.** Probe position.

*5.4. Surgical Site Infiltration*

The last step of anesthesia is wound infiltration, which is usually a transverse surgical incision respecting Langer lines. A transverse incision that is about 4 to 5 cm long is kept 2 to 3 cm above the inguinal ligaments just lateral to midline. After planning the surgical incision, we completed the local wound infiltration with 20 mL of anesthetic (Mepivacaine 10 mg/mL–Ropivacaine 7.5 mg/mL). The surgical site infiltration starts with the infiltration of approximately 10 mL of anesthetic along the line of the incision into the subdermic and intradermic tissue. After superficial local anesthetic injection, the deep subcutaneous injection is performed with approximately 10 mL of local anesthetic.

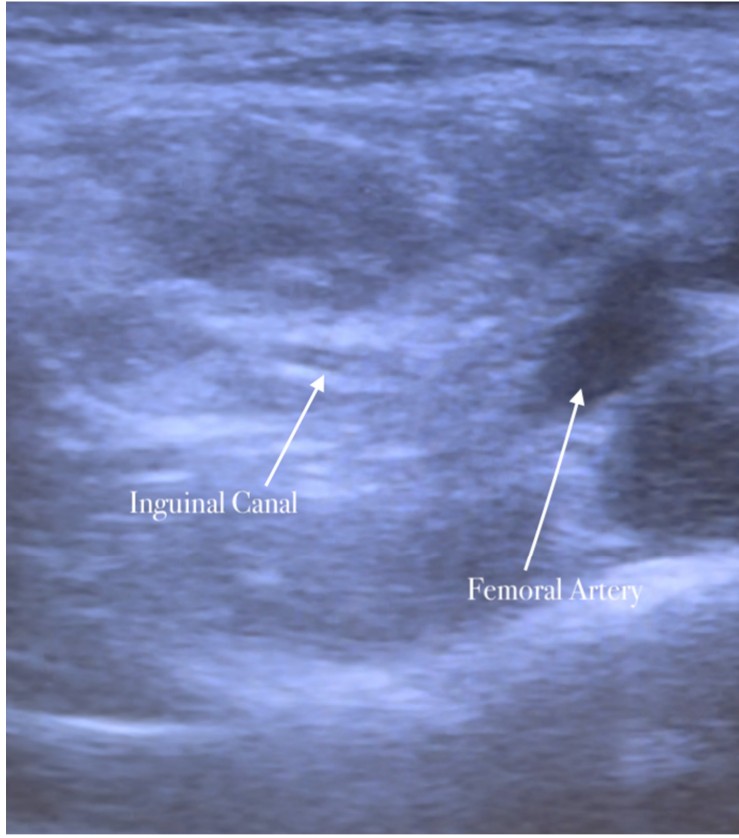

**Figure 6.** Spermatic cord identification.

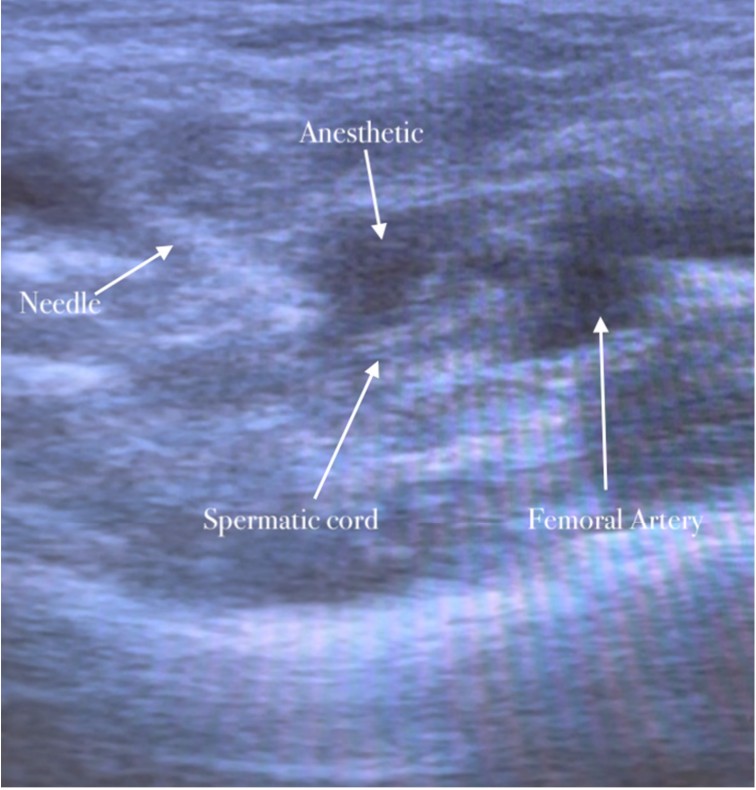

**Figure 7.** The anesthetic is injected in the spermatic cord.

*5.5. Intraoperative Anesthesia*

During surgery, two more injections of local anesthetic are usually performed. After the dissection of the fascia of Scarpa and the exposure of the aponeurosis of the external oblique muscle, 2–3 mL of anesthetic mixture is injected immediately underneath the aponeurosis of the external oblique. It separates the external oblique aponeurosis from the underlying ilioinguinal nerve, thus decreasing the risk of injuring the nerve during the incision of the external oblique aponeurosis.

Another 2–3 mL of anesthetic mixture is injected to block the genital branch of the genitofemoral nerve that travels along the posterior-medial aspect of the spermatic cord together with the cremasteric vein ("blue line").

Occasionally, especially in large inguinal hernias, an infiltration of a few milliliters of anesthetic at the level of the neck of the hernia sac is required to achieve complete local anesthesia.

## 6. Tips and Tricks

Anesthesia alone may not be adequate to control pain during hernia repair. One of the main causes of pain is traction. Traditionally, adequate surgical exposure has been accomplished with the aid of self-retaining retractors or by extending the incision length. Self-retaining retractors can lead to potential complications, including local tissue ischemia and pain.

To reduce traction during hernia repair, we used the Alexis™ S wound retractor. It provides 360 degrees of atraumatic circumferential retraction. The wound protector is placed after the incision of Scarpa's fascia, exposing the aponeurosis of the abdominal external oblique muscle. With their index finger, the surgeon creates space towards the testicle, and then the surgeon places the Alexis™ S wound retractor, as seen in Figures 8 and 9.

The wound protector provides a significant increase in surgical exposure compared to self-retaining retractors. The wound protector reduces the depth of the operative site, and it is helpful in obese patients. The reduction in traction improves postoperative pain due to the lower compressive force.

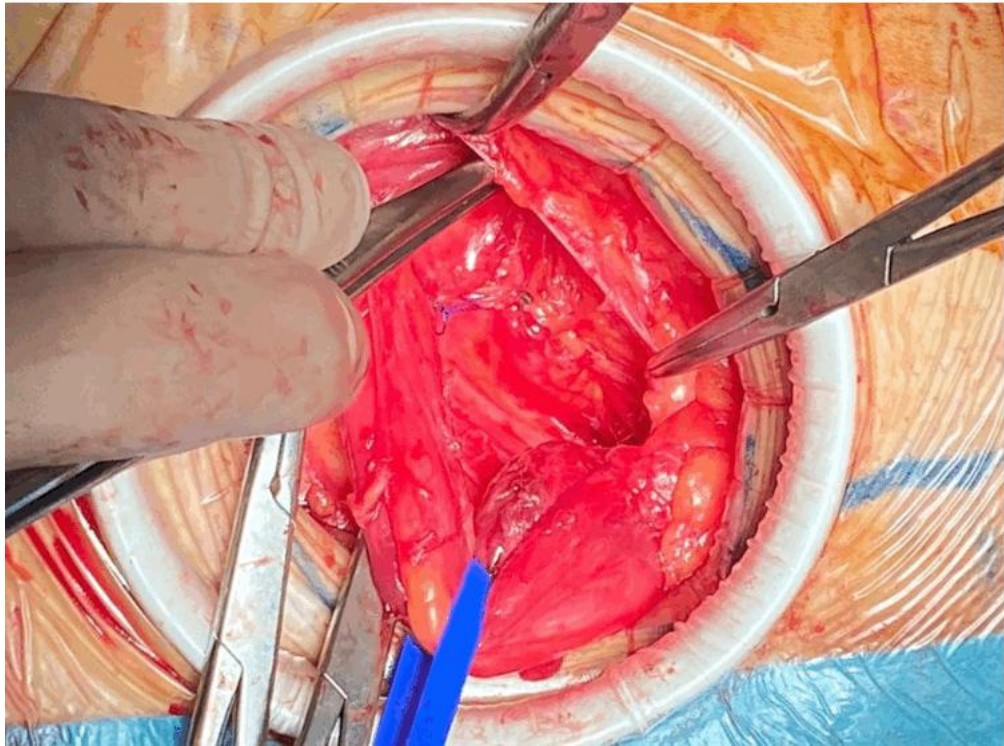

**Figure 8.** Improved surgical exposure using wound retractor.

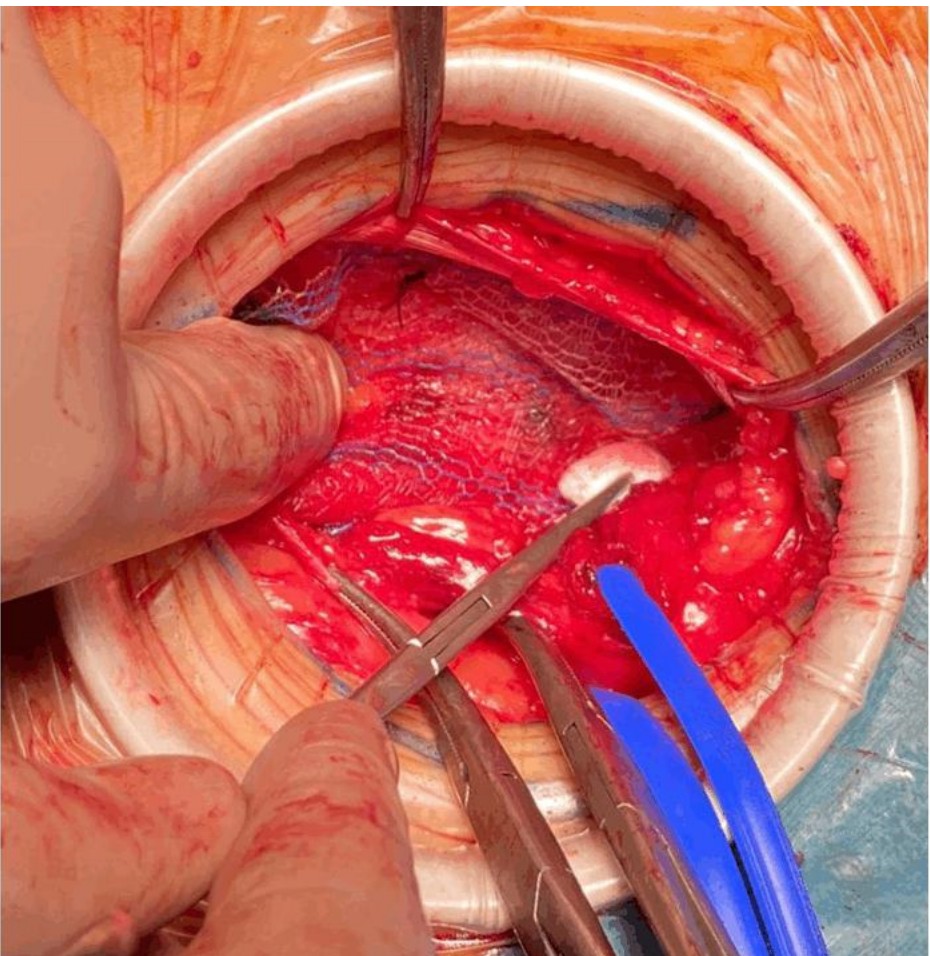

**Figure 9.** Improved surgical exposure using wound retractor during mesh fixation.

Several studies have reported the decreased rates of SSI even in clean wounds such as elective inguinal hernia repair [18–20].

### 7. Pain Control after Surgery

After surgery, 30 mg of Ketorolac is administered. The patient is discharged after complete mobilization 3–5 h after surgery.

Home therapy is usually prescribed for the first seventy-two hours after surgery. The pain treatment consists of 600 mg of ibuprofen two times a day for two days to relieve pain and reduce inflammation, and 1000 mg paracetamol to improve pain control if necessary.

### 8. Results

This protocol for the management of open hernia repair was officially initiated in March, 2022. The protocol was adopted by only one surgeon (FB) in the first months; subsequently, other surgeons became actively involved in the project (MG and GEN).

A total of 408 patients who underwent open groin hernia repair were enrolled in the protocol.

Every patient received ilioinguinal–iliohypogastric and genitofemoral nerves block.

Every patient met the inclusion criteria.

Among the 408 patients included in this study, 276 (67.6%) who were on the waiting list for inguinal hernia repair under spinal or general anesthesia, were evaluated again. After the new clinical evaluation, they were included in the protocol.

After the administration of the ilioinguinal–iliohypogastric and genitofemoral nerves block, no adverse events were reported.

Good analgesic effect was observed in 371 (91.0%) patients. Only 37 (9.0%) patients complained about pain during surgery and required the addition of fentanyl or midazolam to local anesthesia.

After the administration of fentanyl and midazolam before skin incision, only 3/408 (0.7%) patients required the infusion of atropine or flumazenil due to the onset of bradycardia and/or hypotension resulting in the prompt resolution of the medical issue.

An anesthesiologist has never been involved for complication due to the anesthetic administration.

Rescue administration of analgesics during the first 60 min after surgery was necessary in 2% of the patients.

The application of the protocol has led to a sensible reduction in initially planned spinal anesthesia.

Every hernia (100%) repair was made uneventful.

Every patient was treated in day hospital treatment, no one requested prolonged hospitalization or modification of pain relief therapy. No complications, such as intestinal perforation, allergic reactions, postoperative nausea and vomiting, and urinary retention, were reported in the included patients.

No differences among patients initially planned for spinal anesthesia and patients planned initially with nerves block were found.

## 9. Discussion

The lack of anesthesiologists and the need to promptly treat every patient were the main reasons behind the adoption of this protocol.

We evaluated every patient on the waiting list again to apply the protocol. Only patients who met the inclusion criteria were included.

Inguinal hernia repair is a burden disease and also a common surgical procedure, which can be performed under general, spinal or local anesthesia. Inguinal hernia repair is usually performed in the presence of an anesthesiologist. The presence of an anesthesiologist is mandatory for general and spinal anesthesia. General and spinal anesthesia are related to a high risk of complications after surgery, such as postoperative nausea, vomiting, and urinary retention, when compared to local anesthesia [8,16,21].

Ultrasound-guided peripheral nerves block is a good alternative to more invasive spinal anesthesia.

Our results show that only 0.7% of the included patients required the infusion of atropine or flumazenil due to anesthetic complications, and only 37 (9.0%) patients required the addition of fentanyl or midazolam to local anesthesia to improve the intraoperative analgesia effect.

Percutaneous ilioinguinal–iliohypogastric and genitofemoral nerves block is not novel, and it is well described in several papers [7–10,21].

Our results are stackable to other previous studies on percutaneous ilioinguinal–iliohypogastric and genitofemoral nerves block [11–13].

Song et al. [22] conducted a study comparing general spinal anestesia and ilioinguinal–iliohypogastric nerves block. He showed that the VAS in the recovery room 30 min postoperatively was lower in the ilioinguinal–iliohypogastric nerves block group.

Hu et al. [23] conducted a study to evaluate ultrasound-guided ilioinguinal–iliohypogastric nerves block. The study showed that all patients had successful blocks without complications.

Bang et al. [24] conducted a study comparing spinal anesthesia and ilioinguinal–iliohypogastric nerves block showing that patients' satisfaction in the recovery room was similar between the two groups.

The technique for nerves block is safe and easily reproducible. This method offers several advantages when compared to pure local, spinal, or general anesthesia. Ultrasound guidance has dramatically improved the safety and success rate of nerve blockades when compared to the blind infiltration of local anesthetic through different layers due to the risk of inadvertent femoral nerve block and intestinal puncture [14].

Percutaneous ilioinguinal–iliohypogastric and genitofemoral nerves block show a better pain control during and after surgery than local anesthesia and a faster postoperative course when compared to spinal or general anesthesia. These advantages are very important in managing a burden disease that generally requires several resources [16–19].

The wound protector plays an important role in the good results of this method. The wound protector provides a significant increase in surgical exposure compared to self-retaining retractors minimizing traction, and consequently, the postoperative pain. The wound protector also reduces the depth of the operative site, becoming a fundamental tool in obese patients [18–20,25].

Our experience suggests that peripheral nerve blockades can be safely performed in selected patients.

The combination of percutaneous ilioinguinal–iliohypogastric and genitofemoral nerves block in combination with the wound protector in selected patients lead to a good analgesic result during surgery and the postoperative period.

In our experience, the absence of anesthesiologist during inguinal hernia repair is not related to increased risk for the patient. The application of this protocol has not led to a lower pain control or to higher anesthetic infusion complications.

This method seems promising, especially considering the fact that there is a shortage in physicians. Future analysis of the results of this method through the years could lead to a new but old concept to treat inguinal hernia.

Future results could make this method a real game changer in the fight against inguinal hernia.

## 10. Conclusions

The results of percutaneous ilioinguinal–iliohypogastric genitofemoral nerves block in combination with a wound protector suggest, in the present study, the effectiveness of this protocol, showing that inguinal hernia repair can be performed safely even without the presence of an anesthesiologist in the operative room, avoiding the waste of resources. This protocol can be safely replicated everywhere with good results in terms of pain control, anesthetic complications, and the need for the addition of anesthetics addition to local anesthesia.

**Author Contributions:** Conceptualization, M.G. and F.B.; methodology, M.G.; validation, G.E.N., M.G. and F.B.; formal analysis, M.G, G.E.N.; data curation, M.G.; writing—original draft preparation, M.G.; writing—review and editing, M.G.; visualization, G.E.N.; supervision, F.B. All authors have read and agreed to the published version of the manuscript.

**Funding:** This research received no external funding.

**Institutional Review Board Statement:** Ethical review and approval were waived for this study due to the nature of the study. In the study aren't reported any data about patients.

**Informed Consent Statement:** Written informed consent has been obtained from the patient(s) to publish this paper.

**Data Availability Statement:** The original dataset generated during the current study is available from the corresponding author on reasonable request.

**Conflicts of Interest:** The authors declare no conflict of interest.

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
