# Peer review of "Old but Gold: The Surgeon’s Affair to Manage Inguinal Hernia"

_2038-9582, doi:10.3390/std12020008_

Round 1
Reviewer 1 Report
The manuscript is mainly a ”cook-book” for performing open inguinal hernia repair in local anesthesia. The ultrasonographic guided approach in applying local anesthesia and the use of a special retractor are the main novelties. The illustrations are very informative and of high qualitive.
The results are very brief. A total of 408 patients have been operated without conversion to general anesthesia, and only 0.7% needed supplementary medical treatment for minor complications. There is no information on follow-up or any patient related outcome.
The main argument for introducing the method was shortage of anesthesiologists and other staffing. In the introduction it is said that the procedure can be performed with a single nurse assistance, but the present set-up recommends the attendance of two nurses and two surgeons. The authors must comment on this.
To my opinion it is not possible to conclude that this method anesthesia should be superior to other methods in inguinal hernia repair. This also includes the use the wound retractor. Only properly performed randomized studies will give the correct answer on these aspects.
Author Response
Dear Reviewer thank you very much to spend time with our work.
We really appreciated your comments. I hope that you will find this version more clear and easy to read.
In the introduction it is reported that local anesthesia can be performed generally with a single nurse assistance, we recommend the attendance of two nurses and two surgeons on the operative theatre, 1 nurse anesthesiologist and 1 scrub nurse.
We improved results and discussion.
We believe that result section should capture only the anesthetic course due to the nature of the paper where we didn’t observed severe complications in the procedure application
In the discussion we confirmed the genuineness of the method with all the limitation of the study.
Reviewer 2 Report
line 44. inguinal hernia repair is not often accompanied by acute to chronic pain. The vast majority of patients do well, and i dont know how pain relates to the development of cardiovascular events, wound healing?
line 51 - "lack of" anesthesiologists, remove "pain relief"
it is unclear what line 183 means. - "The complications traditionally related to spinal anesthesia have not 183 been reported in 80% of the patients." does this mean that their patients had up to 20% complication rate related to anesthesia administration prior to this protocol being instituted? seems oddly high, and that are the complications they are referring to?
effective use of self retaining retractors can produce the necessary surgical field exposure; claiming a "significant" improvement in exposure and reduction in pain control using the alexis retractor over standard retractors (selfretaining or hand held) is a very subjective and hard to examine result.
Nerve block for open inguial hernia repair is not a novel technique. it has been previously described. In addition, authors do not compare other reported series using nerve blocks in open inguinal hernia repair.
there is merit in adapting to this surgical unit's lack of anesthesiologist coverage, and instead of limiting the amount of patients offered surgical treatment, creating a working protocol to overcome this deficiency in a successful way is commendable. however, if authors claim novelty in this approach compared to other published reports they should provide / discuss those differences in the discussion and make the reader understand the reason why their approach is better. if the point of the manuscript is to document success in managing open inguinal hernia repair under limited resources, then the manuscript achieves that, though the concept is not novel (nerve block to avoid general anesthesia or MAC)
Author Response
Dear Reviewer thank you very much to spend time with our work.
We really appreciated your comments. I hope that you will find this version more clear and easy to read.
We modified the text according to your suggestions
We included other studies on nerves block for inguinal hernia reapir.
Wound retractor's role in exposure and reduction of pain due to lower traction of traditional retractors has been reported in different studies. I'm sorry but we have forgot to include the correct reference in the previous version.
Reviewer 3 Report
Does the use of a dedicated wound retractor to improve site exposure produce a significant rise in the cost of the procedure?
A comparison between the patients with local anesthesia versus those who received spinal anesthesia should be interesting.
Author Response
Dear Reviewers thank you very much to spend time with our work.
We really appreciated your comments. I hope that you will find this version more clear and easy to read.
The wound retractor cost is 23 euro, we believe that it is a reasonable cost to improve the procedure’s outcome.
A comparison between the patients with local anesthesia versus those who received spinal anesthesia should be interesting but unfortunately we have difficulties to recover the data from patients who underwent local or spinal anesthesia.
Round 2
Reviewer 1 Report
All my previous concerns have been adressed sufficiently. The manuscript has improved considerably by focusing upon the method with local anaesthesia and not the surgical results